# Overestimation of Relative Risk and Prevalence Ratio: Misuse of Logistic Modeling

**DOI:** 10.3390/diagnostics12112851

**Published:** 2022-11-17

**Authors:** Charalambos Gnardellis, Venetia Notara, Maria Papadakaki, Vasilis Gialamas, Joannes Chliaoutakis

**Affiliations:** 1Department of Fisheries and Aquaculture, School of Agricultural Sciences, University of Patras, 26504 Patra, Greece; 2Department of Public and Community Health, School of public Health, University of West Attica, 11521 Athens, Greece; 3Department of Social Work, Faculty of Health Sciences, Hellenic Mediterranean University, 71004 Heraklion, Greece; 4Department of Early Childhood Education, School of Education National and Kapodistrian, University of Athens, 10680 Athens, Greece

**Keywords:** etiological studies, relative risk, logistic regression, modified Poisson regression

## Abstract

The extensive use of logistic regression models in analytical epidemiology as well as in randomized clinical trials, often creates inflated estimates of the relative risk (RR). Particularly, in cases where a binary outcome has a high or moderate incidence in the studied population (>10%), the bias in assessing the relative risk may be very high. Meta-analysis studies have estimated that about 40% of the relative risk estimates in prospective investigations, through binary logistic models, lead to extensive bias of the population parameters. The problem of risk inflation also appears in cross-sectional studies with binary outcomes, where the parameter of interest is the prevalence ratio. As an alternative to the use of logistic regression models in both longitudinal and cross-sectional studies, the modified Poisson regression model is proposed.

## 1. Introduction

Often in analytic epidemiology, the main interest of an investigation focuses on the possible causal relationship between a risk factor and the occurrence of a disease (or another binary outcome). The examination of the relationship between the factor and the disease can be conducted both in prospective and retrospective studies. In prospective studies, the association between the factor exposure and the disease is assessed using the relative risk or risk ratio (RR). Relative risk is defined as the ratio of the probability of the disease (outcome) in those exposed to the factor to the probability of the disease in the unexposed. Therefore, as a measure, RR estimates how much higher (or lower) is the chance of developing the disease for those exposed to the risk factor compared to the unexposed. The data of such an investigation can be summarized in a 2-by-2 table of the following form (Table 1):

The RR in Table 1 is estimated by the equation
$$RR=\frac{a/\left(a+b\right)}{c/\left(c+d\right)}=\frac{a\left(c+d\right)}{c\left(a+b\right)}$$

In cases where the etiology of rare diseases (or rare events) is being studied, or generally, in cases where the values of *a* and *c* are very small compared to those of *b* and *d*, the RR can be approximated by the odds ratio (OR).

The OR is defined as the odds of the disease occurrence between exposed and unexposed persons
$$OR=\frac{\left(a/\left(a+b\right)\right)/\left(b/\left(a+b\right)\right)}{\left(c/\left(c+d\right)\right)/\left(d/\left(c+d\right)\right)}=\frac{a/b}{c/d}=\frac{a\cdot d}{b\cdot c}$$

OR is a measure of the strength of the association between a binary factor and an outcome [1,2,3]. When the probability of the outcome (or the incidence of a disease) is low, the difference between the odds ratio and relative risk is negligible.
$$RR=\frac{a\left(c+d\right)}{c\left(a+b\right)}\approx \frac{a\cdot d}{b\cdot c}$$

Theoretically, both relative risk and odds ratio can take any positive value.

In case-control studies, the same definitions can be applied, except that RR cannot be assessed directly. In these investigations, the proportion of cases in the entire population at risk is unknown; therefore, one cannot measure the incidence of the outcome or disease. The assessment of relative risk in case-control studies is performed through OR, which adequately approximates RR, as long as the incidence of the disease is low (rare diseases) [2,3,4].

In cross-sectional investigations, the above definitions are modified and adapted to the cross-sectional concept of prevalence. In these studies, as a measure of the relationship between exposure and disease, the prevalence ratio (PR) is commonly used. That is, the prevalence of the disease in those exposed to a risk factor to the prevalence in the unexposed. The prevalence ratio summarizes the concept of risk in cross-sectional studies but without having the potential causality that exists in longitudinal investigations [2,3]. With the notation presented in Table 1, the prevalence ratio is defined as
$$PR=\frac{a/\left(a+b\right)}{c/\left(c+d\right)}$$

In the case of rare diseases, the ratio is adequately approximated by the prevalence odds ratio (POR)
$$POR=\frac{\left(a/\left(a+b\right)\right)/\left(b/\left(a+b\right)\right)}{\left(c/\left(c+d\right)\right)/\left(d/\left(c+d\right)\right)}=\frac{a/b}{c/d}=\frac{a\cdot d}{b\cdot c}$$

The odds ratio, as a cross-product ratio in a 2-by-2 table, has the property of “reciprocity”, which means changing the reference category of the binary outcome will yield reciprocal estimates [5,6,7,8,9,10]. Its assessment as a point estimate of the corresponding population ratio, as well as the assessment of its confidence intervals, are independent of the reference category of the outcome variable. Furthermore, it is known (from sampling theory) that the logarithmic transformation of OR, in either the form $\frac{a\cdot d}{b\cdot c}$ or the reciprocal $\frac{\;b\cdot c\;}{a\cdot d}$, is normally distributed with estimated standard error, in both cases, equal to $\sqrt{\frac{1}{a}+\frac{1}{b}+\frac{1}{c}+\frac{1}{d}}$. These properties make the logarithm of OR the natural measure (time-invariant) in stochastic-risk modeling [2,3].

It must be emphasized that reciprocity does not apply to the assessment of RR because its definition depends on the reference category of the outcome variable. In the case where RR is defined in terms of the occurrence of the disease, it is estimated by the quantity $\frac{a/\left(a+b\right)}{c/\left(c+d\right)},\;$ while in the opposite case (non-occurrence) by the quantity $\frac{\;b/\left(a+b\right)}{d/\left(c+d\right)}$ (Table 1). Therefore, the interpretation of a comparison based on RR is critically important on whether the positive outcome or its negative complement is modeled [5,6,7].

In contrast to the odds ratio, relative risk has a notable arithmetic property called “collapsibility”. This means that a crude (unadjusted) risk ratio will not change if adjusted for a variable that is not a confounder. Due to collapsibility, assuming no confounding, the RR estimates the change in risk, on a ratio scale, for the entire exposed group due to exposure. Odds ratios are not collapsible, and therefore, the interpretation of an OR is more limited. The odds ratio will estimate the average change in odds for exposed individuals only if all individual odds ratios are the same and all individual risks without exposure are the same. With the exception of this unlikely situation, the crude OR will be closer to 1 than the average of stratum-specific or individual ORs. Even in the absence of confounding, the adjusted OR will be further from 1 than the unadjusted OR [11,12,13,14,15,16].

Odds ratio and relative risk are connected by the relationship $OR=\frac{1-\left(\frac{c}{c+d}\right)}{1-\left(\frac{a}{a+b}\right)}\cdot RR.$ This arithmetic relation indicates that OR tends to overestimate RR by the factor $\frac{1-\left(\frac{c}{c+d}\right)}{1-\left(\frac{a}{a+b}\right)}$ when it is more than 1 or underestimates it when it is less than 1 [17]. However, in cases where the incidence or prevalence of a disease is low (≤10%) OR approximates RR adequately [5]. The overestimation or underestimation of RR by OR tends to be further inflated when it is controlled for potential confounders, as it usually results in logistic regression. A simple method has been proposed to approximate the relative risk from an adjusted odds ratio (obtained from logistic regression) and derive an estimate of an association effect that better represents the true RR [17,18]. However, this method usually produces ratio estimates biased away from 1 when outcomes are common and risk among those not exposed varies substantially, suggesting that the strength of association is greater than is true [19,20]. Moreover, the proposed confidence interval for the adjusted RR, computed by applying the method to the bounds of the adjusted OR’s confidence interval, can also be biased, leading one to believe that the relative risk estimate is more precise than is true. Options exist to obtain unbiased estimates of relative risks in studies of common outcomes. As we will see below, there are methods that have widely available user-friendly software and are statistically appropriate to handle binary common outcomes.

It is common, in statistical analyses of a binary outcome, to use logistic regression as a predominant technique in assessing the risk of exposure or the benefit of a treatment [21,22]. Logistic regression is a linear model with a dependent variable, the log odds of a binary outcome
$$\ln\left(\frac{p}{1-p}\right)=B_{0}+B_{1}x_{1}+B_{2}x_{2}+\ldots +B_{k}x_{k}$$

The basic advantage of logistic regression is the ability to yield odds ratio estimators adjusted for the effect of covariates. However, in cases where the incidence of a disease (or an outcome) is relatively high, the use of ORs frequently yields inflated estimates of the RRs [5,11,13,23,24,25,26].

Modified Poisson regression is an alternative to logistic regression, where the parameters are RRs rather than ORs. It is a generalized linear model with a linking function, the logarithm of the probability of a binary event (instead of the log odds) [27]. For the data summarized in Table 1, the probability of the disease in those exposed to the risk factor is defined by
$$p=\frac{a}{a+b}$$

A simple linear model, with outcome variable, logarithm of *p* and predictor the exposure variable *X*, is expressed by the equation
(1)$$\ln\left(p\right)=B_{0}+B_{1}x$$

For *x* = 1 (exposed) and *x* = 0 (unexposed) the values of (1) are
(2)$$\ln\left(p_{\left(x=1\right)}\right)=B_{0}+B_{1}\cdot 1=B_{0}+B_{1}\;$$
(3)$$\ln\left(p_{\left(x=0\right)}\right)=B_{0}+B_{1}\cdot 0=B_{0}\;$$

The exponential transformation of (2) and (3) gives
(4)$$p_{\left(x=1\right)}=e^{\left(B_{0}+B_{1}\right)}$$
and
(5)$$p_{\left(x=0\right)}=e^{B_{0}}$$

RR is defined as the ratio of the probability of the outcome of the exposed (*x* = 1) divided by the probability of the unexposed (*x* = 0). Using Equations (4) and (5), RR is finally defined by
$$RR=\frac{p_{\left(x=1\right)}}{p_{\left(x=0\right)}}=\frac{e^{\left(B_{0}+B_{1}\right)}}{e^{B_{0}}}=\frac{e^{B_{0}}\cdot e^{B_{1}}}{e^{B_{0}}}=e^{B_{1}}$$

If a Poisson distribution is assumed, the model parameters are estimated by a maximum likelihood estimation algorithm (quasi-likelihood method, QL) [28]. Estimating the coefficients of the simple model results in
$$e^{B_{0}}=\frac{c}{c+d}\;\mathrm{and}\;e^{B_{1}}=\frac{\alpha \left(c+d\right)}{c\left(\alpha +b\right)}$$

The estimated standard error of RR, after being corrected for data overdispersion (using robust estimates for the covariance matrix), is given by [29]
$$\sqrt{\frac{1}{a}-\frac{1}{a+b}+\frac{1}{c}-\frac{1}{c+d}}$$

In the case of multiple predictor variables, the simple Poisson model can be extended by the equation
(6)$$\ln\left(p\right)=B_{0}+B_{1}x_{1}+B_{2}x_{2}+\ldots +B_{k}x_{k}$$
or, equivalently
(7)$$p=e^{B_{0}+B_{1}x_{1}+B_{2}x_{2}+\ldots +B_{k}x_{k}}\;,\;$$
where *x*_1_, *x*_2_,…*x_k_* are the values of a set of predictor variables *Χ*_1_, *Χ*_2_,…, *Χ*_k_.

The interpretation of the model (6) is similar to the multiple logistic regression one, with the difference that its coefficients are referred to the probability (and not to the odds) of an outcome. Specifically, for each variable *Χ_i_*, the quantity $e^{B_{i}}$ is the factor by which the probability of an outcome is multiplied when the independent variable *Χ_i_* increases by one unit (given that the other variables remain constant).

It should be mentioned that the above use of Poisson’s regression is an exceptional use of Poisson’s model, which is commonly applied to count data. This specific model application, with the error variance correction, is generally referred to as modified Poisson regression or robust Poisson regression [27].

## 2. Materials and Methods

In a clinical study, survival of mixed lobular and ductal carcinomas of breast cancer was compared with a series of patients with pure invasive ductal ones. One hundred and thirty-two patients, ninety-eight with ductal carcinomas and thirty-four with mixed who had been treated by modified radical mastectomy, were followed for 10 years [26]. Survival data were available for all patients with a minimum of five-year follow-up period. At the end of follow-up time, their survivorship was assessed in relation to the histological type of tumor. The data are presented in Table 2.

The OR and RR of five-year mortality for the mixed versus ductal type were, respectively:$$OR=\frac{a\cdot d}{b\cdot c}=\frac{26\cdot 60}{8\cdot 38}=5.13,\;$$
$$RR=\frac{a\left(c+d\right)}{c\left(a+b\right)}=\frac{26\cdot 98}{38\cdot 34}=1.97$$

Moreover, the 95% Confidence Intervals (CIs) of OR and RR were: 95% CΙ OR (2.11, 12.50) and 95% CI RR (1.45, 2.69).

## 3. Results

Using a simple logistic regression model, with the five-year mortality as a binary outcome (dead/alive) and the histological type as a predictor, the estimated OR for mixed versus ductal carcinoma mortality at the end of five-year period is the same as in the simple 2-by-2 table, that is, about 2.5 times the RR (Table 3).

Furthermore, using a multiple logistic regression model controlling for age, grade of malignancy (Grade II, Grade III), possible lymphocytic infiltration, and the number of lymph nodes metastases (≤3, >3), the estimated OR is further inflated to 5.25 (Table 4).

The results of logistic model indicate that adjusted OR tends to overestimate RR, even to a greater extent than unadjusted one. Moreover, simulation studies have shown that estimated CIs for the OR are quite wide, while the point estimation of the ratio appears to have inconsistency problems (that is, the bias does not decrease as the sample size increases) [19,23,29]. When estimating RR of an outcome, the use of logistic regression model is considered abusive if the outcome incidence is rare (<10%). For common outcomes, OR always overstates RR, sometimes dramatically. In these cases, the modified Poisson regression model has been proposed [29,30,31].

The modified Poisson regression model will be used to reanalyze the survival data of breast cancer. The results of the modified Poisson regression, with dependent variable the five-year mortality and predictor the histological type, are presented in Table 3. The table includes the regression-derived coefficient (B), the estimation of RR, the Wald test, and the CI for RR. The point and interval estimation of RR are identical to those calculated directly by the contingency table (RR = 1.97, 95% CI: 1.45–2.69).

In order to adjust for covariates, the survival data were reanalyzed, controlling for histological type, degree of malignancy, possible lymphocytic infiltration, number of metastatic lymph nodes, and age (Table 4).

The adjusted RR for mixed versus ductal adenocarcinoma five-year mortality still remains about twice as high (RR = 1.94, *p* < 0.001). Moreover, the CIs for all predictor variables are robust and consistent compared to those obtained from the logistic regression model. Specifically, the 95% CI for the histological type is (1.36, 2.75) versus (1.94, 14.22) of the logistic model.

It is evident that the assessment of RR in Poisson regression is more coherent compared to the logistic model. The difference from its unadjusted assessment in the 2-by-2 contingency table between the histological type and the present situation is negligible (1.94 vs. 1.97), implying that its adjustment for potential confounding variables is small. Furthermore, the introduction of multiple independent variables in the Poisson model yields more efficient coefficient estimators than the logistic model (more compact CIs).

The estimation of RR through the Poisson model is performed in terms of a specific outcome (death in our example) without the “reciprocity” that appears in logistic model. That is, while in logistic regression, regardless of the outcome against which the risk is estimated, the exponentiated coefficients are reciprocal of each other (with the significance remaining the same), in the Poisson model, both the coefficients and statistical significance are modified according to the estimated outcome. This differentiation may seem strange to those familiar with the use of logistic regression, but it is consistent with the rationale of Poisson model since the dependent variable in the latter (i.e., the logarithm of the relative risk) varies according to the outcome of interest. In extreme cases, there may be statistical significance for one category of the binary outcome and not for the other [5].

The breast cancer data will be reanalyzed to show the differences in the symmetry (reciprocity) between the logistic regression and the Poisson model results. In the new analysis, the binary outcome will be survival and not death (Table 5). Exponentiated coefficients of logistic model (that is, RRs) are reciprocal of the original ones (with death as a reference event), while in the Poisson regression, they differ in both values and statistical significance. For example, the adjusted RR for survival of mixed adenocarcinoma versus ductal is OR = 0.19 = 1/5.25, *p* = 0.001, of Grade III versus Grade II, OR = 0.65 = 1/1.55, *p* = 0.360, etc. On the contrary, reciprocity does not apply to the Poisson regression coefficients: for mixed adenocarcinoma vs. ductal, RR = 0.40 ≠ 1/1.94 = 0.52, *p* = 0.005, for Grade III vs. Grade II, RR = 0.79 ≠ 1/1.18 = 0.85, *p* = 0.353, etc.

## 4. Discussion

Binary logistic regression is widely used in empirical research, sometimes even uncritically. Based on the functional form of the logistic model, it results that the exponentiated coefficient of a dichotomous independent variable (defining two groups) estimates the OR of the binary outcome. If the investigation simply regards the existence of a significant relationship between the independent variable and the outcome with no interest in unbiased estimation of RR, logistic regression is extremely useful with extensive applications.

However, if the focus is on estimating the RR of a binary outcome, then the use of OR, as derived from the exponentiated coefficients of the logistic regression model, is unsafe. This estimate is very close to the RR when the outcome has a low incidence in the studied data (≤10%). Nevertheless, in cases where the incidence is higher, the use of logistic regression tends to overestimate the RR and sometimes misleadingly inflate it. In prospective epidemiological studies, when exposure to a risk factor is evaluated, the logistic regression model should be used with great caution. In these cases, where the estimation of RR can be performed directly from the data, it is suggested to use the modified Poisson model as described previously. Especially in cases of common outcomes with a frequency higher than 10% in the study data, the use of the modified Poisson regression is considered necessary. The same procedure is proposed for the analyses of randomized clinical trials [12]. Meta-analysis studies have estimated that approximately 40% of the RR assessment in prospective investigations through logistic regression models leads to extensive biases of the corresponding population parameters [32].

The binary logistic model can be used in a reliable and consistent manner if the outcome has a low incidence in the sample data. It can also be used in studies with the main interest in predicting an outcome or estimating the strength and direction of the association between the independent variables and the dichotomous outcome. In these latter cases, where there is no need for unbiased estimation of RR, logistic regression works satisfactorily with information-rich output. Furthermore, in case-control epidemiological studies, the use of logistic regression is the only option since the RR cannot be directly estimated from the available data. In these specific investigations, the only possible option is to estimate the ORs.

In cross-sectional studies, the use of logistic regression is subject to the same limitations as in longitudinal studies [5,33,34,35]. In these cases, since we are interested in unbiasedly estimating the RR of an outcome, the use of the modified Poisson model is suggested (since the outcome has a low incidence in the available data). If the purpose of a cross-sectional study is not to estimate the relationship (in terms of strength and direction) of a dichotomous variable and an outcome but to unbiasedly estimate the prevalence ratio in two groups, then the aforementioned limitations are equally applied.

Apart from the modified Poisson regression, as an alternative procedure for estimating relative risk, the log binomial model has been proposed and is widely used. The latter has similar power to the robust Poisson regression (possibly, in some cases, providing slightly smaller standard errors of the relative risk) but often exhibits convergence problems of the maximum likelihood algorithm it uses [36,37]. Moreover, it generates incorrect estimates when the covariate values in the data are not bounded by one (that is, the case of continuous covariates) [36]. Results similar to those of Poisson regression are also provided by the Cox proportional hazard model with robust variance estimation. The Cox model is equivalent to Poisson regression if the follow-up time for all subjects in the study is considered constant (in the analysis, we can set the follow-up time to one or any quantity that is the same for all subjects). Another simple alternative for risk estimation is to use the Mantel–Haenszel method for relative risk, which can adjust for one or two confounders [38,39]. However, this method becomes inefficient when several factors, especially continuous variables, are being adjusted for simultaneously. The literature is rich regarding the effectiveness and unbiasedness of all previous methods. Considering the weaknesses and strengths of the methods, as well as their ease of use and availability in commercial data analysis software, Poisson regression seems to have the most advantages [31,40,41,42]. Nevertheless, researchers should keep in mind that due to the lack of symmetry in Poisson regression coefficients (as with all models with the relative risk as a dependent variable), the choice of reference category is critically important in whether the positive outcome or its negative complement is modeled. Moreover, the study results and inference should not be based solely on statistical significance but also on clinical significance, which sometimes is overlooked, and undue importance to statistical measures is given.

## 5. Conclusions

The uncritical application of logistic regression in epidemiological studies often creates inflated estimates of the RR. The basic advantage of logistic regression is the ability to yield approximations of the RRs through odds ratios adjusted for the effect of possible covariates. However, in cases where the incidence of a disease or an outcome is relatively high, ORs can substantially overestimate RRs. An alternative to the use of logistic regression is the modified Poisson model. As a basic criterion for choosing the appropriate method, it is suggested primarily to construct a 2-by-2 contingency table between the risk factor and the outcome and then calculate both the OR and the RR. If the two values differ notably, the use of the modified Poisson model is recommended. Since this procedure is implemented in standard statistical packages, there is no justification for relying on logistic regression when the relative risk is the parameter of primary interest.

In a future paper, validation of the performance of modified Poisson regression, in terms of relative bias for point estimation and percentage of confidence interval coverage, will be undertaken through simulated and real data sets.

## Figures and Tables

**Table 1 diagnostics-12-02851-t001:** Notation for entries in a 2-by-2 table.

	Outcome (Disease)		
Factor	Event	No Event	Total
exposed	*a*	*b*	*a + b*
unexposed	*c*	*d*	*c + d*
Total	*a + c*	*b + d*	*n = a + b + c + d*

**Table 2 diagnostics-12-02851-t002:** Survivorship of 132 women at the end of five-year follow-up period.

		Dead	Alive	Total
Histological type	Mixed	26	8	34
		76.5%	23.5%	100.0%
	Ductal	38	60	98
		38.8%	61.2%	100.0%
Total		64	68	132
		48.5%	51.5%	100.0%

**Table 3 diagnostics-12-02851-t003:** Simple logistic and Poisson regression models with five-year mortality as the outcome variable.

**Logistic Regression**	* **Β** *	* **OR** *	***p*-Value**	**95% CI *OR***
Histological type (mixed vs. ductal)	1.64	5.13	<0.001	2.11–12.50
**Poisson Regression**	** *Β* **	** *RR* **	***p*-Value**	**95% CI *RR***
Histological type (mixed vs. ductal)	0.68	1.97	<0.001	1.45–2.69

**Table 4 diagnostics-12-02851-t004:** Multiple logistic and Poisson regression models with five-year mortality as the outcome variable.

**Logistic Regression**	** *Β* **	** *OR* **	***p*-Value**	**95% CI *OR***
Histological type (mixed vs. ductal)	1.66	5.25	<0.001	1.94–14.22
Age of patients (in years)	0.01	1.01	0.723	0.95–1.08
Grade of malignancy (Grade III vs. Grade II)	0.44	1.55	0.360	0.61–3.95
Lymphocytic infiltration	0.66	1.93	0.219	0.68–5.51
Lymph node metastases (>3)	0.32	1.38	0.430	0.62–3.10
**Poisson Regression**	** *B* **	** *RR* **	***p*-** **Value**	**95% CI *RR***
Histological type (mixed vs. ductal)	0.66	1.94	<0.001	1.36–2.75
Age of patients (in years)	0.01	1.01	0.688	0.98–1.03
Grade of malignancy (Grade III vs. Grade II)	0.17	1.18	0.380	0.81–1.72
Lymphocytic infiltration	0.29	1.34	0.250	0.81–2.22
Lymph node metastases (>3)	0.16	1.17	0.368	0.82–1.68

**Table 5 diagnostics-12-02851-t005:** Multiple logistic and Poisson regression models with five-year survival as the outcome variable.

**Logistic Regression**	** *Β* **	** *OR* **	***p*-Value**	**95% CI *OR***
Histological type (mixed vs. ductal)	−1.66	0.19	0.001	0.07–0.52
Age of patients (in years)	−0.01	0.99	0.723	0.93–1.06
Grade of malignancy (Grade III vs. Grade II)	−0.44	0.65	0.360	0.25–1.65
Lymphocytic infiltration	−0.66	0.52	0.219	0.18–1.48
Lymph node metastases (>3)	−0.32	0.72	0.430	0.32–1.62
**Poisson Regression**	** *B* **	** *RR* **	***p*- ** **Value**	**95% CI *RR***
Histological type (mixed vs. ductal)	−0.92	0.40	0.005	0.21–0.76
Age of patients (in years)	−0.01	0.99	0.659	0.97–1.02
Grade of malignancy (Grade III vs. Grade II)	−0.24	0.79	0.353	0.48–1.30
Lymphocytic infiltration	−0.24	0.78	0.208	0.54–1.15
Lymph node metastases (>3)	−0.12	0.88	0.434	0.65–1.20

## Data Availability

Data are available on request.

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
