# Peer review of "Overestimation of Relative Risk and Prevalence Ratio: Misuse of Logistic Modeling"

_diagnostics, 2022, doi:10.3390/diagnostics12112851_

Round 1
Reviewer 1 Report
This paper presents an attractive topic, and it needs some improvements.
The introduction needs further improvement in terms of clarity, the organization of this paper also week. The mathmatical notations of this paper are not clear almost in the whole article.
Experiments to compare with state-of-the-art methods can be used.
Statistical analysis can also be further discussed.
Some related works should be used:
Add numbering to the used equations
I suggest the authors to use the template of the journal
The conclusion is short and you should extend it with some future works
Author Response
Replies to Reviewer #1 comments
This paper presents an attractive topic, and it needs some improvements.
Reply We would like to thank the Reviewer very much for the time spent on our work, as well as the useful comments made that helped us, we believe, to improve the presentation of our findings.
Comment 1: The introduction needs further improvement in terms of clarity, the organization of this paper also week. The mathematical notations of this paper are not clear almost in the whole article.
Reply 1: The authors tried to address all the above comments. The introduction has been improved and made clearer. Mathematical definitions were also improved and numbering was added to equations for better understanding by non-mathematical readers.
Comment 2: Experiments to compare with state-of-the-art methods can be used.
Reply 2: As we say in the conclusion of the paper, validation of the performance of Poisson regression, in terms of relative bias (for point estimation and confidence interval coverage) as well as comparisons with other state-of-the-art methods, will be undertaken in a future work.
Comment 3: Statistical analysis can also be further discussed.
Reply 3: With the reconstruction of the article according to the journal template and the additions made, both the reported statistical methods for the analysis of binary data and the analyzes made on the breast cancer example, were expanded.
Comment 4: Some related works should be used: Add numbering to the used equations
Reply 4: Numbering was added in the revised ms, as proposed.
Comment 5: I suggest the authors to use the template of the journal
Reply 5: The template of the Journal was used in the revised ms.
Comment 6: The conclusion is short and you should extend it with some future works
Reply 6: The conclusion was extended in the revised ms, as proposed.
Reviewer 2 Report
Here, Gnardellis and colleagues analyse the use of prevalence ratio and relative risk in epidemiological studies and highlight the excessive use of logistic regression which may lead to erroneous, false-positive results by generating inflative values of these parameters. According to their estimates, around 40% of the relative risk calculations in prospective investigations and prevalence ratio calculations in cross-sectional studies made using binary logistic regression creates a bias, thereby skewing the results. Instead, they suggest the use of the modified Poisson regression, a generalized linear model form of regression analysis used to model count data and contingency tables.
As an example, the authors provide calculations where simple logistic and Poisson regression models are employed for the evaluation of 5-year mortality from breast cancer as the outcome variable, showing that odds ratio is 5.13 when using simple logistic regression and relative risk is 1.97 when applying a modified Poisson regression. Likewise, similar were the odds rations and relative risks for the comparison of 5-year mortality depending on histological type of breast cancer, age of patients, grade of malignancy, lymphocytic infiltration, and lymph node metastases when performing multiple logistic regression and modified Poisson regression (5.25 and 1.94 for histological type; 1.55 and 1.18 for grade of malignancy, 1.93 and 1.34 for lymphocytivc infiltration, 1.38 and 1.17 for lymph node metastases, and 1.01/1.01 for age of patients).
The evaluation of 5-year survival rates by multiple logistic and Poisson regression models demonstrated the corresponding results (OR = 0.19 and RR = 0.40 for histological type; 0.65 and 0.79 for grade of malignancy; 0.52 and 0.78 for lymphocytic infiltration; 0.72 and 0.88 for lymph node metastases; 0.99 and 0.99 for age of patients).
To me, the results are clear and highlight the potential usefulness of Poisson regression instead of simple or multiple logistic regression for the precise calculation of odds ratio/relative risks in epidemiological studies.
Author Response
Replies to Reviewer #2 comments
Here, Gnardellis and colleagues analyse the use of prevalence ratio and relative risk in epidemiological studies and highlight the excessive use of logistic regression which may lead to erroneous, false-positive results by generating inflative values of these parameters. According to their estimates, around 40% of the relative risk calculations in prospective investigations and prevalence ratio calculations in cross-sectional studies made using binary logistic regression creates a bias, thereby skewing the results. Instead, they suggest the use of the modified Poisson regression, a generalized linear model form of regression analysis used to model count data and contingency tables.
As an example, the authors provide calculations where simple logistic and Poisson regression models are employed for the evaluation of 5-year mortality from breast cancer as the outcome variable, showing that odds ratio is 5.13 when using simple logistic regression and relative risk is 1.97 when applying a modified Poisson regression. Likewise, similar were the odds rations and relative risks for the comparison of 5-year mortality depending on histological type of breast cancer, age of patients, grade of malignancy, lymphocytic infiltration, and lymph node metastases when performing multiple logistic regression and modified Poisson regression (5.25 and 1.94 for histological type; 1.55 and 1.18 for grade of malignancy, 1.93 and 1.34 for lymphocytivc infiltration, 1.38 and 1.17 for lymph node metastases, and 1.01/1.01 for age of patients).
The evaluation of 5-year survival rates by multiple logistic and Poisson regression models demonstrated the corresponding results (OR = 0.19 and RR = 0.40 for histological type; 0.65 and 0.79 for grade of malignancy; 0.52 and 0.78 for lymphocytic infiltration; 0.72 and 0.88 for lymph node metastases; 0.99 and 0.99 for age of patients).
To me, the results are clear and highlight the potential usefulness of Poisson regression instead of simple or multiple logistic regression for the precise calculation of odds ratio/relative risks in epidemiological studies.
Reply: We would like to thank the Reviewer very much for the time spent on our work as well as the useful comments regarding the clarity and the usefulness of the certain paper.
Reviewer 3 Report
The paper is very interesting and well written. It is acceptable for publication in the current form
Author Response
Replies to Reviewer #3 comments
The paper is very interesting and well written. It is acceptable for publication in the current form.
Reply: We would like to thank the Reviewer very much for the time spent on our work as well as the useful comments.
Round 2
Reviewer 1 Report
accept